# The Epitranscriptome in miRNAs: Crosstalk, Detection, and Function in Cancer

**DOI:** 10.3390/genes13071289

**Published:** 2022-07-21

**Authors:** Daniel del Valle-Morales, Patricia Le, Michela Saviana, Giulia Romano, Giovanni Nigita, Patrick Nana-Sinkam, Mario Acunzo

**Affiliations:** 1Division of Pulmonary Diseases and Critical Care Medicine, Department of Internal Medicine, Virginia Commonwealth University, Richmond, VA 23298, USA; delvallemod@vcu.edu (D.d.V.-M.); patricia.le@vcuhealth.org (P.L.); michela.saviana@vcuhealth.org (M.S.); giulia.romano@vcuhealth.org (G.R.); patrick.nana-sinkam@vcuhealth.org (P.N.-S.); 2Comprehensive Cancer Center, Department of Cancer Biology and Genetics, The Ohio State University, Columbus, OH 43210, USA; giovanni.nigita@osumc.edu

**Keywords:** miRNA, epitranscriptomics, cancer

## Abstract

The epitranscriptome encompasses all post-transcriptional modifications that occur on RNAs. These modifications can alter the function and regulation of their RNA targets, which, if dysregulated, result in various diseases and cancers. As with other RNAs, miRNAs are highly modified by epitranscriptomic modifications such as m^6^A methylation, 2′-O-methylation, m^5^C methylation, m^7^G methylation, polyuridine, and A-to-I editing. miRNAs are a class of small non-coding RNAs that regulates gene expression at the post-transcriptional level. miRNAs have gathered high clinical interest due to their role in disease, development, and cancer progression. Epitranscriptomic modifications alter the targeting, regulation, and biogenesis of miRNAs, increasing the complexity of miRNA regulation. In addition, emerging studies have revealed crosstalk between these modifications. In this review, we will summarize the epitranscriptomic modifications—focusing on those relevant to miRNAs—examine the recent crosstalk between these modifications, and give a perspective on how this crosstalk expands the complexity of miRNA biology.

## 1. Introduction

The central dogma depicts a straightforward transfer of genetic information from DNA to RNA to proteins [1]. However, the cell represents a dynamic environment in which these biological macromolecules can be modified through myriad processing events. The epitranscriptome specifically encompasses the post-transcriptional modifications of coding and non-coding RNAs such as piwi-interacting RNAs (piRNAs), transfer RNAs (tRNAs), ribosomal RNA (rRNA), and microRNAs (miRNAs) [2].

Many RNA processing events have been characterized within cells, including N6-methyladenosine (m^6^A), 2′-O-methyl, 5-methylcytidine (m^5^C), 7-Methylguanosine (m^7^G), poly-uridine (poly-U), and adenosine-to-inosine (A-to-I) nucleotide substitutions [3]. These post-transcriptional modifications display a fundamental impact on cell physiology, such as the development of the nervous system [4], stem cell differentiation [5], circadian clock regulation [6], and heat shock response [7].

Unclear for decades, part of the functional significance of these RNA modifications has been recently uncovered. These modifications play a fundamental role in mRNA decay, splicing, alternative polyadenylation, export, stabilization, and translation [8], and perturbations to these finely-tuned events have even been associated with a wide range of diseases, including the progression of human cancers [3].

Similar to other RNAs, miRNAs are modified with epitranscriptomic modifications that can alter their functionality, regulation, and biogenesis [9]. miRNAs are short non-coding RNAs (sncRNAs) that are master regulators of gene expression at the post-transcriptional level. Since their initial discovery, thousands of miRNAs have been identified throughout eukaryotes, some with a high degree of sequence conservation [10]. miRNAs are first transcribed by RNA polymerase II (pol II), which binds either to dedicated miRNA gene loci or intronic regions of known protein-coding genes and transcribes a primary miRNA (pri-miRNA) [11]. The pri-miRNA transcript is subsequently processed through a series of cleavages to produce the mature miRNA [11,12]. While still in the nucleus, the long pri-miRNA transcript is trimmed into an intermediate hairpin structure (precursor miRNA/pre-miRNA) by the Microprocessor, a complex composed of the RNase III enzyme Drosha and DiGeorge syndrome critical region 8 (DGCR8) binding protein [11,12]. The pre-miRNA is exported from the nucleus, through the nuclear transport receptor exportin-5, to the cytoplasm, where it is cleaved by the RNase III enzyme Dicer to generate the resultant mature miRNA duplex [11,12]. This duplex is ~21–22 nucleotides long with short 3′ overhangs (~2 nucleotides) [11,12]. Shortly after that, a single “guide” strand of the miRNA duplex is loaded into an effector complex alongside Argonaute (AGO) proteins to form an RNA-induced silencing complex (RISC) [13]. The guide strand possesses a seven-nucleotide-long “seed sequence” in its 5′ region that base-pairs with complementary sequences within the 3′ untranslated regions (UTRs) of target mRNAs [11,14,15]. The formed RNA duplexes invoke the degradation or translational repression of the target mRNA, leading to eventual gene silencing [14]. A single miRNA can regulate the expression of hundreds or thousands of targets by this mechanism of RNA interference (RNAi) [11]. Due to the wide range of gene targets, miRNAs are studied extensively in relation to diseases such as cancer [15]. miRNAs are involved in many hallmarks of cancer, and their expression is related to tumor development and progression [15]. Based on their function and dysregulation in malignancies, miRNAs are classified as oncomiRNAs, which target onco-suppressor genes, and tumor suppressor miRNAs, which target oncogenes, impeding their downstream functions [16]. Epitranscriptomic modifications of these miRNAs lead to alteration in their targeting [9]. Given that, the number of miRNA targets becomes much more complex when considering epitranscriptomic modifications, as both the miRNA and the targeted gene can be modified [9].

The most common epitranscriptomic modifications found in miRNAs are m^6^A, A-to-I editing, 2′-O-methyl, and m^5^C, although other modifications have been reported including m^7^G and poly-U [9]. Moreover, crosstalk between m^6^A and other epitranscriptomic modifications has recently been described throughout the literature [17]. This crosstalk aims to regulate the effects of the epitranscriptomic modifications, adding a layer of complexity to the post-transcriptional regulation of gene expression.

This review will briefly describe the role of epitranscriptomic modifications in RNA biology and disease, focusing on those involved in miRNA biology and the most used methods to detect them. We will also describe the recently reported crosstalk between these modifications, highlighting their implications for miRNA biology.

## 2. The Epitranscriptome in RNA

To date, over 150 unique epitranscriptomic modifications have been identified in RNAs [18]. These modifications play a key role in altering the function and regulation of both coding and non-coding RNAs. m^6^A, A-to-I, m^5^C, m^7^G, poly-U, and 2′-O-methyl have been reported in miRNAs. These modifications play a key role in miRNA processing and downstream mRNA targeting, as shown in Figure 1. As such, the epitranscriptome wholly represents an added layer of complexity in gene regulation.

### 2.1. N6-Methyladenosine (m^6^A)

m^6^A was initially characterized in the 1970s as a prevalent modification found in mRNAs, accounting for approximately 50% of methylated ribonucleotides [19,20]. Research interest in m^6^A skyrocketed in recent years due to high throughput sequencing techniques coupled with m^6^A RNA immunoprecipitation that facilitated the identification of m^6^A sites [21]. m^6^A is a dynamic and reversible modification enriched at the 3′UTR and near the stop codons of mRNAs [22]. m^6^A has been detected in other sites of the mRNA such as introns to promote alternative splicing [23], the CDS to affect translation elongation rate [24], and the 5′UTR to promote cap-independent translation [25,26]. The m^6^A writers METTL3/METTL14 [27], METTL5 [28], METTL16 [29,30], WTAP [31], VIRMA [32,33], ZC3H13 [34], HAKAI [35], and RBM15 [36] methylate adenosines, as well as m^6^A can be removed by the m^6^A erasers FTO [37] and ALKBH5 [38]. Recently, FTO has been shown to predominantly demethylate N6,2′-O-dimethyladenosine (m^6^Am) over other m^6^A sites [39]. m^6^Am is the combined methylation of m^6^A and 2′-O-methly that occurs adjacent to the 5′ cap site of mRNAs. FTO can demethylate m^6^Am and render the mRNA susceptible to decapping and degradation [39]. m^6^A readers (YTHDF 1/2/3, YTHDC1/2, IGF2BP1/2/3, EIF3, PRRC2A, HNRNPA2B1, and HNRNPC/G) [40] bind to the m^6^A site and can alter the stability of mRNAs, as well as promote protein translation, splicing, and mRNA export [41]. m^6^A can target a wide range of mRNA targets, leading to the regulation of various cellular processes. Dysregulation of m^6^A is associated with various diseases [42] such as neuronal disorders [43], osteoporosis [44], metabolic disease [45], and various cancers [46]. METTL3 and METTL14 play important physiological roles in mammalians, such as stem cell differentiation and reprogramming [47,48,49], or impacting the circadian cycle [6]. Dorn et al. showed that m^6^A modification controls cardiac homeostasis, as it is improved in response to hypertrophic stimuli and is required for the normal response of cardiomyocytes [50]. m^6^A is also important in brain development; in fact, METTL3 silencing provokes severe movement disorders and brain dysplasia [51].

In cancer, m^6^A writers, erasers, and readers disrupt the regulation of oncogenes and tumor suppressors, leading to an increase in cancer cell proliferation, tumor growth, and migration [46]. Based on the type of tumor and its targets, the enzymes involved in m^6^A modification can induce tumor-suppressive or oncogenic pathways with different effects and have been proposed as a potential therapeutic target. For instance, METTL3 is upregulated in acute myeloid leukemia (AML) and is a necessary gene to preserve the undifferentiated phenotype in AML [52]. In lung cancer, METTL3 promotes proliferation by methylating Bcl-2 mRNA and increasing its translation [53].

#### m^6^A and miRNAs

In miRNAs, m^6^A is found in pri-miRNAs, where it enhances the recognition and binding of DGCR8 to promote miRNA processing [54]. RNA immunoprecipitation with an anti-m^6^A in HEK293 cells identified 239 enriched miRNAs; 20.9% of all miRNAs contain the consensus METTL3 motif RRACH [55]. Knockdown of METTL3 led to a global downregulation of mature miRNAs and an accumulation of unprocessed pri-miRNAs [54]. Likewise, METTL14 regulates the processing of miR-375 [56]. Overexpression of METTL14 in colorectal cancer suppresses cell growth via miR-375 targeting YAP1 [56]. Knockdown of the m^6^A eraser FTO altered the steady-state levels of miRNAs [57]. A recent study shows that METTL14 is decreased in hepatocellular carcinoma (HCC) and is associated with metastasis in vivo and in vitro [58]. Moreover, METTL14 regulates the processing of the tumor suppressor miR-126 in an m^6^A-dependent manner, and the downregulation of METTL14 in HCC results in a reduction of miR-126 [58].

The m^6^A enzymes are also targets of miRNAs [59], which in turn have cancer-specific expression. For example, miR-33a [60] and miR-4429 [61] target METTL3 mRNA, modulating global m^6^A levels in non-small-cell lung cancer (NSCLC) and gastric cancer, respectively. miR-145 targets the m^6^A reader YTHDF2; their expression is inversely correlated in ovarian cancer cells [62]. Furthermore, FTO is targeted by miR-1266; knockdown of miR-1266 promotes cell proliferation in colorectal cancer [63]. Taken together, m^6^A modifications increase miRNA processing, while miRNAs themselves can target m^6^A enzymes.

### 2.2. 2′-O-Methyl

2′-O-methyl is a modification commonly found in rRNAs [64], animal piRNAs [65], and mRNAs [66]. The 2′-O-methylation of RNAs does not alter Watson–Crick base pairing but instead stabilizes the nucleotide conformation, restricting the rotational freedom of the 3′ phosphate [67]. These properties protect the RNA from hydrolytic cleavage [67]. In rRNA, 2′-O-methyl is a post-transcriptional modification added by a ribonucleoprotein complex containing the C/D box of small nucleolar RNA (snoRNA) and the methyltransferase fibrillarin [68]. The 2′-O-methylation of rRNA contributes to the structural stabilization of rRNA, forming hydrophobic interactions between nucleotides [69]. Interestingly, knockdown of fibrillarin decreases the translation efficiency of genes containing internal ribosome entry sites (IRES), suggesting a subset of mRNAs that can be differentially regulated by 2′-O-methyl-containing ribosomes [70].

In mRNA, 2′-O-methyl can be found either internally within the gene body [71] or at the first nucleotides as part of the 5′ cap structure [72]. The 5′ end of RNAs transcribed by RNA pol II is modified by an m^7^-guanosine cap. The first and second nucleotide of the 5′ end of pre-mRNAs can be further methylated with 2′-O-methyl, forming the cap1 and cap2 structures, respectively [72]. The cap1 structure is the predominant cap found in mammalian cells [73], where the 2′-O-methyl acts as a specific marker for self-made mRNAs [74]. mRNAs that lack a 2′-O-methyl at the first nucleotide are recognized by RIG-I, triggering an innate immune response [75].

Furthermore, 2′-O-methyl has been detected internally in mRNAs [71]; its biological role was recently elucidated. Internal 2′-O-methyl increases the mRNA abundance of the peroxidase PXDN while reducing its protein translation [66]. 2′-O-methyl sites occurring in codons block interactions between rRNA and the codon-anticodon helix, hindering translation elongation [76].

#### 2′-O-Methyl in miRNAs

2′-O-methylation of miRNAs has been primarily reported in plant and drosophila miRNAs [77,78]. In plants, HEN1 adds the 2′-O-methyl and is essential for the biogenesis of miRNAs [77]. The 2′-O-methylation promotes the stabilization of plant miRNAs by protecting the 3′ end of miRNAs from truncation and degradation [77]. Knockdown of HEN1 resulted in heterogeneous 3′ ends and the addition of poly-U, a modification known to destabilize RNAs (discussed later in this review) [79]. In *Drosophila*, 2′-O-methyl was detected in specific isoforms of miRNAs [78]. 2′-O-methylation of miRNAs and their loading with AGO2 increases with age. Mutations of HEN1 induced neurodegeneration and decreased lifespan in *Drosophila*, suggesting miRNA 2′-O-methyl impacts aging in *Drosophila* [78].

Recently, 2′-O-methylation of miRNAs was detected in NSCLC and paired distal non-cancerous lung tissues, with a differential 2′-O-methyl status of miRNAs in both tissues [80]. HENMT1 was identified as the methyltransferase responsible for 2′-O-methylation in human miRNAs [80]. 2′-O-methylation of miR-21-5p was found to be increased in NSCLC. Collectively, this methylation enhanced the resistance of miR-21-5p to 3′ degradation by PNPT1, enhanced the affinity of miR-21-5p to AGO2 loading, and enhanced the repression of the miR-21-5p target gene, PDCD4 [80]. This study suggests that 2′-O-methylation of miRNAs can increase the repressive activity of miRNAs by preventing their degradation from enzymes such as PNTP1.

### 2.3. 5-Methylcytosine (m^5^C)

m^5^C was discovered almost 50 years ago [81], and yet its role has only recently been uncovered. RNA m^5^C methylation is a dynamic and reversible process driven by several factors, described as writers, readers, and erasers [82]. The RNA m^5^C methyltransferases (RCMTs) are mainly represented by the NOL1/NOP2/SUN domain (NSUN) family and DNA methyltransferase 2 (DNMT2), which transfer the methyl group from the donor S-adenosylmethionine to the cytosine, forming m^5^C [83].

The NSUN family of proteins comprises seven members, named NSUN1 to NSUN7 [82]. While NSUN1, NSUN2, and NSUN5 are conserved throughout eukaryotes, the remaining NSUN proteins are only present in higher eukaryotes [84]. NSUN1 and NSUN5 methylate cytoplasmic rRNAs, NSUN2 and NSUN6 modify cytoplasmic tRNA, and NSUN3 and NSUN4 methylate mitochondrial RNAs [84]. NSUN2 is a tRNA m^5^C methyltransferase identified as a mediator of m^5^C in mRNA and non-coding RNA [85]. The mRNA export adaptor protein ALYREF (Aly/REF export factor) recognizes m^5^C methylation in mRNA through a methyl-specific RNA-binding motif and mediates the export of m^5^C-containing RNA [86,87]. Additionally, NSUN2 modulates ALYREF’s nuclear–cytoplasmic shuttling.

DNA methyltransferase 2 (DNMT2) is a methyltransferase that catalyzes m^5^C in the 38th cytosine of tRNAs [88]. DNMT2 methylates tRNA^Asp^_GUC_ and, depending on the species, tRNA^Gly^, tRNA^Val^, and tRNA^Glu^ [89]. C38 methylation of tRNA^Asp^_GUC_ is important for the amino acid charging of the tRNA [90]. Knockdown of DNMT2 in mice reduced the amount of charged tRNA^Asp^_GUC_ and reduced the synthesis of proteins containing poly-Asp [90].

Currently, the approaches used for detecting m^5^C modifications in RNA include bisulfite sequencing, m^5^C-RIP-seq, Aza-IP-seq, and miCLIP-seq, which are discussed later in the review (reviewed in [82]). In 2012, Squires et al. combined bisulfite treatment with next-generation sequencing of a cellular RNA library to map novel candidate m^5^C sites in several types of cellular RNA [91]. The distribution of m^5^C methylation occurred in tRNA and mRNA sites but also untranslated regions. In particular, an enrichment of m^5^C sites was observed in the UTR of mRNAs and proximity of the AGO binding site, suggesting a role in the post-transcriptional control of RNA functions.

m^5^C methylation affects mRNA transport, stability, and translation [82]. Additionally, m^5^C methylation of tRNA prevents its degradation from oxidative stress [92]. The distribution and abundance of m^5^C in several types of RNA are critical for the physiological function of the cell, and its deregulation is related to pathological features [82]. Mutations in the NSUN family of proteins are associated with autosomal-recessive intellectual disability for NSUN2 [93], mitochondrial deficiency for NSUN3 [94], and sterility for NSUN7 [95]. NSUN2 has been associated with promoting cell progression, growth, and metastasis in various cancers [96]. NSUN2 can target the 3′UTR of hepatoma-derived growth factor (HGDF) mRNA and increase its stability, promoting *urothelial carcinoma pathogenesis* [97].

#### m^5^C in miRNAs

m^5^C methylation has been recently reported as present in microRNAs. Using a non-targeted mass spectrometry sequencing technique, Konno et al. identified methylated cytosines in mature sequences of miR-21-3p and miR-200c-3p in gastrointestinal cancer cells [98].

The mechanisms involved in cytosine methylation have only recently been elucidated. In a pivotal study, Cheray et al. demonstrated that the complex DNMT3A/AGO4 is responsible for the cytosine methylation of miR-181a-5p [99]. m^5^C on miRNA inhibits the formation of miRNA/mRNA duplexes, thus impairing the miRNA’s ability to bind to the target mRNA. miR-181a-5p is frequently under-expressed in glioblastoma [100], and its role as a tumor suppressor has only recently been discovered [101]. The removal of m^5^C from miR-181a-5p abolishes its tumor-suppressive function [101].

Interestingly, the cytosine methylation of miR-181a-5p was also associated with a poor prognosis of glioblastoma, which indicates the novel usefulness of cytosine-methylated miRNAs as a prognostic biomarker for cancer [99]. A significant fraction of miRNAs was identified as containing m^5^C [99]. However, further investigation is needed to identify the regulatory mechanism of m^5^C in other miRNAs.

### 2.4. 7-Methylguanosine (m^7^G)

m^7^G is an abundant modification found in the 5′ guanosine cap of mRNAs, as well as internally in mRNAs, tRNAs, rRNAs, and miRNAs [102]. m^7^G is critical for the complete maturation of the 5′ guanosine cap found in all RNAs transcribed by RNA pol II [103]. In the nucleus during RNA pol II transcription, the enzymes RNA Guanylyltransferase and 5′-Phosphatase (RNGTT), RNA Guanine-7 Methyltransferase (RNMT), and RNA Guanine-7 Methyltransferase Activating Subunit (RAMAC) associate with the phosphorylated C-terminal domain of RNA pol II. RNGTT guanylates the 5′ end of the RNA, and RNMT-RAMAC adds m^7^G to the cap, forming the cap0 structure [103]. RNMT-RAMAC also participates in the methylation of recapped mRNAs in the cytoplasm [104,105]. Proper methylation of the cap is required for eIF4E binding for translation initiation and for the stability of the mRNA. Improperly methylated capped mRNAs are targeted for degradation by the cap surveillance enzymes DXO/Dom3Z [106].

The enzymatic activity of RNMT-RAMAC is regulated during embryonic stem cell differentiation [107]. ERK1/2 phosphorylates RAMAC, targeting it for degradation and modulating the cap methylation of pluripotency-associated genes [107]. Elevated protein levels of RNMT and RNGTT have been observed in high-eIF4E acute myeloid leukemia patients; elevated capping levels of MALAT, RNMT, and MYC were observed in these patients [108]. Downregulation of RNMT selectively inhibits the proliferation of PIK3CA mutant breast cancer cell lines, suggesting that cap methylation is required for PIK3CA mutated cancer cells [109].

METTL1 and WDR4 catalyze m^7^G in various RNAs. METTL1/WDR4 modifies m^7^G at the G46 site of a tRNA variable loop, stabilizing the tertiary structure of tRNAs [110]. Disruption of G46 methylation in tRNAs is associated with microcephalic primordial dwarfism [110]. m^7^G was recently detected internally in mRNAs. Internal m^7^G is catalyzed by METTL1/WRD4 and is associated with increased translation efficiency [111]. In rRNA, m^7^G is catalyzed by the WBSCR22/TRMT112 complex, where G1639 in 18S rRNA is methylated. This methylation participates in the biogenesis of 18S rRNA [112,113].

#### 7-m^7^G in miRNAs

The presence of m^7^G modification in the 5′ guanosine cap of mRNAs has been widely studied, yet there are few papers researching this in miRNAs. In their effort, Xie et al. developed a small RNA Cap-seq protocol to identify a group of m^7^G-capped pre-miRNAs, including a subset whose 5′ ends coincide with the RNA pol II transcription start site (TSS) [114,115]. These m^7^G-capped pre-miRNAs undergo a non-canonical biogenesis pathway that bypasses Drosha and exportin-5 and instead goes through PHAX-dependent exportin-1 [114,115]. The identified pre-miRNAs possess 5′-terminal extensions which do not impede Dicer processing [115]. In fact, rather than being trimmed by Dicer to canonical miRNAs, the generated 5p miRNAs retain the 5′ extension; still, it is the 3p miRNAs that are preferentially loaded in AGO [114,115]. In light of this finding, the researchers proposed a novel strategy in which an RNA pol II promoter could be positioned to result in m^7^G-capping of shRNAs and the subsequent selection of a single 3p siRNA for targeted silencing [114]. Intriguingly, Kaur et al. found that CD47 indirectly interacts with exportin-1 via ubiquilin-1, its known cytoplasmic binding partner, and limits the intracellular trafficking of m^7^G-capped miRNAs and mRNAs into extracellular vesicles [116].

Martinez et al. observed a subset of miRNAs whose expression was induced in response to quiescence despite a marked reduction of exportin-5 [117]. The quiescence-induced miRNAs were processed in an exportin-1-dependent manner, but instead of being strictly m^7^G-capped, they have detectable trimethylguanosine (m2,2,7G/TMG) caps that are added at the pri-miRNA stage [117]. A reduction in the levels of TMG-capped pri-miRNAs was observed after a knockdown of TGS1, an enzyme that catalyzes the hypermethylation of the 5′ cap [117]. Kamel and Akusjärvi reported that, after initiation at the adenovirus major promoter, RNA pol II stalling/termination produces an m^7^G-capped TSS small RNA (sRNA) transcript [118]. In human adenovirus-infected cells, this m^7^G-capped TSS sRNA is enriched in AGO2-containing RISC, unlike the aforementioned 5′-extended 5p miRNAs, and is capable of repressing complementary pTP and Adpol mRNAs, consequently suppressing viral DNA replication [118].

Kouzarides et al. used two complementary techniques, RNA immunoprecipitation sequencing (RIP-seq) and borohydride reduction sequencing (BoRed-seq), to identify a high-confidence group of mature miRNAs that contain the m^7^G modification at internal positions [119]. In a series of experiments, they demonstrate that METTL1 directly methylates pri-let-7e, a tumor suppressor miRNA, and facilitates its processing by disrupting local G-quadruplex structures [119]. In comparison, Vinther et al. implemented a different technique which they called m^7^G mutational profiling sequencing (m^7^G-MaP-seq) but could not find evidence for the m^7^G modification in any human miRNAs, including let-7e [120].

### 2.5. 3′ Poly-Uridine (Poly-U)

Poly-U is an epitranscriptomic modification in which a short string of non-templated uridines is added to the 3′ end of RNAs by Terminal Uridylyl Transferases (TUTases) [121]. Poly-U results in the degradation of the respective RNA or influences maturation of snRNAs and, in some cases, miRNAs [121]. In mammals, TUT4, TUT7, and TUT1 are responsible for the addition of 3′ poly-U to RNAs [122,123,124]. The nuclear TUT1 plays a role in the maturation of U6 snRNA. After transcription of U6 snRNA, TUT1 adds 20 uridines to the 3′ end. This long poly-U tail acts as a signal for further maturation of U6 by USB1 [125]. The cytoplasmic TUT4 and TUT7 add the poly-U tail to other RNAs. As part of the RNA quality control mechanism, defective or improperly matured ncRNAs are tagged with poly-U; the exonuclease DIS3L2 recognizes these poly-U ncRNAs for degradation [126].

In mRNAs, poly-U is added to mRNAs with a short (~25nt) poly-A tail and leads to the degradation of the mRNA [122,127]. Knockdown of TUT4 and TUT7 increased the half-life of various mRNA transcripts [122]. Histone mRNAs, which lack a poly-A tail and instead are stabilized by a stem-loop, require poly-U for their proper turnover during the cell cycle [128]. miRNA-directed cleavage of mRNAs is reported to add a stretch of poly-U downstream of the cleavage site. This poly-U coordinates the decapping and 5′ degradation of the cleaved mRNA [128].

#### Poly-U in miRNAs

In select pre-miRNAs, poly-U blocks Dicer cleavage and marks the pre-miRNA for degradation [129,130]. This mechanism is best characterized in the let-7 miRNA family. Let-7 pre-miRNA is bound by Lin28, which recruits TUT4 to add a 3′ poly-U tail. The poly-U tail is recognized by DIS3L2 to rapidly degrade the pre-let-7 miRNA [131]. Let-7 miRNA targets a wide range of oncogenes such as MYC, RAS, and HMGA2 [132]. As such, Lin28 has been identified as an oncogene whose overexpression reduces levels of let-7 and has been associated with cancer transformation, proliferation, and advanced malignancies [129].

In addition, poly-U has been reported in mature miRNAs [121]. The 3′ end of miR-26 was shown to be uridylated by TUT4, inhibiting miR-26-mediated IL-6 mRNA degradation [133]. The uridylation was shown to be isomer-specific, with isomir miR-26a showing extensive uridylation which was not present in miR-26b. Interestingly, uridylation of miR-26b was reported in naïve CD8 T-cells, suggesting a cell-dependent uridylation of isomirs [133].

### 2.6. A-to-I Editing

It has been several decades since researchers uncovered an RNA editing event in which adenosine residues are converted to inosines, such as the mechanism behind the double-stranded RNA (dsRNA) unwinding observed in *Xenopus laevis* oocytes [134,135]. An enzyme known as Adenosine Deaminase Acting on RNA (ADAR) was discovered to catalyze this adenosine-to-inosine (A-to-I) nucleotide substitution [136]. The ADAR family has three members: ADAR1, ADAR2, and ADAR3. While ADAR1 and ADAR2 are constitutively expressed in the organism, ADAR3 is primarily expressed in the brain [137]. Each contains up to three dsRNA binding domains and a deaminase domain [138]. ADARs catalyze hydrolytic deamination at the C6 position of adenosine, converting it into an inosine residue which is interpreted by the host translational machinery as if it were guanosine [139]. A-to-I editing changes the transcript sequence and influences alternative splicing [140,141]. Additionally, A-to-I editing in the non-coding region can alter the base-pairing properties of secondary structures, as well as regulate the stability and localization of RNA [142].

A-to-I nucleotide conversions of mRNAs can alter the amino acid sequence, thus creating various protein isoforms [139,143]. From an evolutionary point of view, the production of different proteins stimulates the organism’s adaptation to external stresses. An intriguing case is the hydroxytryptamine subtype 2C receptor (5-HT2CR), a serotonin receptor whose pre-mRNA includes five editing sites and can result in different isoforms [144]. In this case, the editing decreases the affinity of the receptor for its G protein, regulating serotonergic signal transduction [145]. The differential pattern of 5-HT2CR editing has been associated with psychiatric disorders [144,146].

Aberrant regulation of ADAR-mediated editing results in many human diseases, such as Aicardi–Goutières syndrome [147,148], neurological disorders [149], and cancer [138]. A-to-I editing is necessary for the correct functioning of the organism. For instance, the functionally critical decoding of glutamine to arginine of the AMPA receptor subunit GluR-B is operated by ADAR2; the larger and positively charged arginine reduces the Ca^2+^ permeability of the AMPA receptor, and it is essential for producing a functional protein. Consequently, deficiency in this single point of editing is lethal in mice [150]. It has been reported that the GluA2 Q/R site is significantly under-edited in glioblastoma multiforme (GBM) tissue samples [151], and Ca^2+^ signaling mediated by the AMPA receptor activates Akt, thus inducing growth and motility in glioblastoma cells [152]. The levels of ADAR1 have been found to be upregulated in HCC tissue, and the pre-mRNA transcripts of the AZIN1 gene, encoding the antizyme inhibitor 1, were found to be hyper-edited [153]. The HCC cells exhibiting the highest percentage of editing increased proliferation.

#### A-to-I Editing of miRNAs

ADARs recognize and bind to dsRNA substrates, including all forms of miRNAs (pri-miRNA, pre-miRNA, and mature miRNAs), with minimal sequence specificity [143]. Based on the MiREDiBase database, about 2900 putative and validated A-to-I editing events in 571 human miRNA molecules have been described [153]. In the nucleus, ADARs can compete with Drosha for pri-miRNAs, thereby suppressing Drosha-mediated cleavage of pri-miRNAs to pre-miRNAs [143,154]. Yang et al. observed that editing of pri-miR-142 by ADAR1/2 interfered with Drosha processing, reducing mature miR-142 levels [155]. Heale et al. [156] similarly reported that ADARs can block the maturation of miR-376a-2, independent of their enzymatic activity, by merely binding to its pri-miRNA. The predominant consequence of A-to-I editing events is the inhibition of miRNA biogenesis, as observed with miR-142, miR-376a-2, miR-221, miR-222, and miR-21 [155,156].

These nucleotide substitutions can also alter miRNAs’ functions [139]. Specific A-to-I edits within the seed sequences of miRNAs can induce a targetome shift [157,158]. For example, Kawahara et al. [157] have demonstrated that editing within the seed regions of miR-376 cluster members leads to new target genes (i.e., phosphoribosyl pyrophosphate synthetase 1). A-to-I editing of miR-589-3p shifted its target from PCDH9, a tumor suppressor, to ADAM12, a metalloproteinase involved in glioblastoma cell invasion [159]. In melanoma cells, Velazquez-Torres et al. [160] have proposed that edited miR-378a-3p, but not its wild-type counterpart, can target the PARVA oncogene and prevent the malignant transformation of these cells. Xu et al. [161] have similarly found that edited miR-379-5p specifically binds to CD97 and that its delivery in vivo leads to suppression in tumor growth in their mouse model of breast cancer. Interestingly, editing events of the miRNA seed region are observed in response to hypoxia, suggesting a rapid adaptation to environmental stimuli [162].

As mentioned previously, the inosine residue is interpreted as guanosine by the host translational machinery [139]. When Kume et al. evaluated the thermodynamic stability of inosine:cytosine (I:C) and guanine:cytosine (G:C) base pairing between miRNAs and target mRNAs, they found that the former was less stable, which may account for discrepancies in the silencing efficiency of edited miRNAs [158]. Databases, such as miR-EdiTar, document the predicted miRNA binding sites that are either affected by or emerging from editing events [163].

In recent years, high-throughput sequencing technologies have uncovered the widespread dysregulation of edited miRNAs across human malignancies, including neurological diseases, infections, and cancer [163]. For example, in cases of moderate-to-severe asthma, Magnaye et al. [164] found that reduced editing of miR-200b-3p may lead to overexpression of its target SOCS1. In cancers, edited miRNAs are correlated with tumor histology, disease stage, prognosis, and survival [165]. Based on sequencing data from The Cancer Genome Atlas (TCGA), Pinto et al. [166] indicate that hypo-editing of miRNAs is observed globally in most cancers. An exception is miR-200b, which is over-edited in thyroid tumors [167]. When Wang et al. [165] analyzed the TCGA data, they saw that editing hotspots were either observed in nearly all cancers or cancer-specific. Consequently, edited miRNAs are promising cancer biomarker candidates. Maemura et al. [168] have found shorter overall survival in lung adenocarcinoma cases with reduced levels of edited miR-99a-5p, which they suggest as a potential biomarker of this disease. Nigita et al. [169] have also shown, for the first time, that edited miR-411-5p is downregulated in both tissues and exosomes of NSCLC patients.

## 3. Crosstalk between Epitranscriptomic Modifications

Each epitranscriptomic modification has its respective function in RNAs. They do not occur in isolation; RNAs can have multiple epitranscriptomic modifications occurring simultaneously throughout the body of the RNA. These epitranscriptomic modifications can establish crosstalk that is either cooperative (m^6^Am [39] and m^6^A/m^5^C [170]) or regulatory, where one modification controls the expression of another (m^6^A and A-to-I editing) [171,172,173] Figure 2. The crosstalks described in this review are centered on m^6^A interacting with other modifications, as these interactions have been the focus of recent studies.

### 3.1. Crosstalk between m^6^A and A-to-I Editing

Crosstalk between m^6^A and A-to-I editing was suggested in [172], where the co-occurrence of both modifications was negatively correlated. A-to-I editing was shown to occur in m^6^A-negative transcripts preferentially. Of note is that suppression of the m^6^A writers METTL3 and METTL14 increased global A-to-I editing levels in HEK293T cells. ADAR1 is predicted to be a target of m^6^A methylation. This prediction was confirmed in [173] and [171]. Contrary to [172], in both studies, ADAR1 mRNA was identified as a target of METTL3, and its m^6^A site is recognized by the m^6^A reader YTHDF1 to upregulate the protein expression of ADAR1. Knockdown of YTHDF1 in interferon-induced cells decreases global editing of RNAs [173]. The difference in the response of ADAR1 to METTL3 knockdown suggests a context-dependent crosstalk between m^6^A and A-to-I editing. As described earlier, miRNAs undergo A-to-I editing, which raises the question of whether METTL3 can modulate A-to-I editing in miRNAs, which could be studied in future investigations.

### 3.2. N6,2′-O-Dimethyladenosine (m^6^Am)

m^6^Am is a combination of two methylation events, 2′-O-methyl and m^6^A. The first nucleotide of a pre-mRNA in mammals contains a 2′-O-methyl in addition to the 5′ methyl-guanosine cap [174]. If the first nucleotide is an adenosine, it can be further methylated at the N6 position, forming an m^6^Am. m^6^Am is highly abundant in mRNAs initiating with adenosine; 92% of mRNAs that begin with adenosine contain m^6^Am in HEK 293 cells [39]. This m^6^A event is generated by a specific methyltransferase, CAPAM, which interacts with RNA pol II and is coordinated with mRNA capping [175,176]. mRNAs with m^6^Am are resistant to mRNA decapping by DCP2 and are thus more stable. Because of the resistance to decapping, m^6^Am mRNAs are also resistant to miRNA downregulation [39]. For instance, when miR-155 is transfected into HeLa cells, the target genes containing m^6^Am are more resistant to miR-155-induced degradation than other mRNAs [39]. The m^6^A modification in m^6^Am can be removed by FTO, acting as a possible on/off switch for miRNA targeting and degradation [39]. Both miR-155 and FTO are dysregulated in cancers [177,178], and the interplays between m^6^Am in miR-155 targets and FTO could be a possible regulatory mechanism exploited in cancers.

### 3.3. Cooperative Interaction between m^6^A and m^5^C

m^6^A modification by METTL3/METTL14 was shown to facilitate the addition of m^5^C by NSUN2 and vice versa, suggesting a cooperative addition of both modifications [170]. In the case of p21 mRNA, m^6^A and m^5^C were shown to enhance its protein translation cooperatively [170]. Certain viruses, such as HIV and the murine leukemia virus, are enriched in both m^6^A and m^5^C [179,180]. Viral infection with HIV was shown to increase the m^6^A and m^5^C statuses of host cells, changing the methylation of host and viral RNAs [180]. m^6^A and m^5^C are enriched at the 3′UTR of mRNAs, close to predicted miRNA binding sites [181]. m^5^C, in particular, is enriched at AGO binding sites [91]. Given the proximity of these modifications to miRNA binding sites, they could interfere with miRNA binding. More investigation is needed to determine this relationship and if these sites are dysregulated in cancer.

## 4. Methods for Detecting Epitranscriptomic Modifications

Epitranscriptomic modifications were initially identified in the 1970s using two-dimensional thin-layer chromatography to identify RNA methylations [81]. Although thin-layer chromatography is still widely used, recent advances in RNA sequencing methods and mass spectrometry have allowed for the transcriptome-wide identification of epitranscriptomic modifications. The techniques used to study epitranscriptomics are described in Table 1.

### 4.1. Thin-Layer Chromatography (TLC)

TLC is a technique that separates chemical compounds in mixtures based on chemical properties such as polarity [182]. The sample is placed on a sheet (stationary phase), and then the sheet is submerged in a light layer of a solvent (mobile phase). The solvent will be absorbed onto the sheet through capillary action, and the sample will migrate with the solvent. Each compound will migrate based on its affinity to the sheet. The separated compounds can be visualized by UV or radioactive labeling of the sample prior to TLC.

A version of TLC that allows for two-dimensional separation (2D-TLC) was used to initially identify RNA methylations present in cells [81]. Typically, the RNA is digested to generate a 5′OH and is radiolabeled with radioactive ATP to visualize the migration pattern of the sample [183]. The migration of the sample RNA is compared with the migration patterns of synthetic RNA standards. By utilizing a standard for each RNA modification, one can identify which epitranscriptomic modification is present in the sample. Additionally, enzyme activity and kinetics can be studied [183]. TLC provides a general overview of the RNA modifications present in the sample RNA. A caveat of TLC is that it will not provide the sequence context or location of the modified nucleotide.

### 4.2. Liquid Chromatography–Mass Spectrometry (LC-MS)

LC-MS is a powerful technique for detecting and quantifying epitranscriptomic modifications in RNA [185]. It combines the sample separation of liquid chromatography (LC) with the mass quantification of mass spectrometry (MS). Prior to analysis, the RNA sample is cleaved into nucleosides and dephosphorylated. The resulting nucleosides are separated into single nucleotides by LC, and their corresponding mass is determined through MS. Each epitranscriptomic modification has its own corresponding mass and isotopic signature, allowing for the detection of known and unknown RNA modifications. LS-MC is highly sensitive, detecting up to femtomole amounts of modified nucleotides [186].

Due to the fragmentation of the RNA sample prior to analysis, LC-MS cannot directly determine the sequence context or identity of the modified nucleotide. To determine the RNA sequence, a ladder with known fragmentation patterns is included in the LC-MS analysis [187]. This limits the sequence analysis to known RNA sequences and modification sites. Recently, new varieties of LC-MS, such as 2-dimensional hydrophobic end-labeling strategy into traditional mass spectrometry-based sequencing (2D HELS MS Seq), have allowed de novo sequencing of modified RNA samples [188].

### 4.3. RNA Sequencing

RNA sequencing combined with immunoprecipitation or chemical modification is one of the most used approaches for profiling the methylation of nucleic acids. Several methods have been developed during past years to detect epitranscriptomic modifications.

#### 4.3.1. Methylated RNA Immunoprecipitation Coupled with High-Throughput Sequencing (MeRIP-seq) and m5C-RIP-seq

A commonly used method for studying the epitranscriptome is methylated RNA immunoprecipitation (RIP) coupled with high-throughput sequencing (MeRIP-seq). This method was applied for the first time in 2012 to study the m^6^A distribution of RNA [22]. In this approach, the purified mRNA is fragmented into 100–150 nucleotides prior to immunoprecipitation with a specific anti-m^6^A antibody which recognizes and enriches RNA fragments carrying the modified nucleotide [22,189]. Subsequently, the RNA fragments are subjected to library construction and deep sequencing. This method is easily manageable and has been adapted for studying m^5^C RNA methylation (m^5^C-RIP-seq) and m^7^G methylation (m^7^G-MeRIP) [111,190]. This transcriptome-wide protocol has high specificity; however, it cannot detect modifications with single-nucleotide resolution and cannot identify the methylation of non-abundant RNA [82].

#### 4.3.2. RNA Crosslinking and Immunoprecipitation (CLIP) Methods

RNA-binding proteins can be covalently linked to RNA through treatment with UV radiation and immunoprecipitated to identify the binding sites of their respective RNA targets. RNA crosslinking and immunoprecipitation (CLIP)-based methods and their derivations have been utilized to identify the sites of epitranscriptomic modifications [220]. The addition of a UV crosslinking step into the meRIP-seq has allowed for single-nucleotide resolution of the methylated nucleotide. The m^6^A-individual nucleotide resolution crosslinking and immunoprecipitation (miCLIP-m^6^A) is an immunoprecipitation-based sequencing method that includes UV crosslinking of the anti-m^6^A bound to the m^6^A site [191]. The crosslinking reaction induces a mutation at the crosslinked m^6^A, allowing for the identification of the exact m^6^A site. An alternative method called photo-crosslinking-assisted m^6^A sequencing (PA-m^6^A-seq) utilizes 4-thiouridine (4SU) that is incorporated into the RNA [193]. The RNA is immunoprecipitated with an anti-m^6^A antibody and crosslinked using UV. Subsequently, the RNA is digested into 25–30 nucleotide fragments and sequenced [194]. Given that 4SU induces a T-to-C mutation at the crosslinking site, T-to-C modifications are identified when compared to the reference genome, allowing methylation detection [195]. Crosslinking can also be used to detect m^5^C. The methylation-individual nucleotide resolution crosslinking and immunoprecipitation (miCLIP) approach can be modified to use an NSUN2 antibody for detecting RNA fragments targeted by NSUN2 [88]. An overexpression of the mutant form of NSUN2 (C271A) will result in a covalently linked RNA-protein complex without the need for UV crosslinking [85]. An overexpression of the mutant form of NSUN2 (C271A) will result in a covalently linked RNA-protein complex after UV crosslinking [83]. Immunoprecipitation is performed using an antibody against NSUN2, and the pulled-down RNA is then used for library construction. Given that the crosslinked covalent bond induces a stop position during RT-PCR, the m^5^C positions are recognized as truncation sites along with the transcriptome [83,192].

#### 4.3.3. m^6^A-Level and Isoform-Characterization Sequencing (m^6^A-LAIC-seq)

m^6^A-level and isoform-characterization sequencing (m^6^A-LAIC-seq) is a high-throughput technique that permits the evaluation of methylation status in the whole transcriptome [194]. An excess of anti-m^6^A antibody is used to ensure the pull-down of methylated RNA, and m^6^A-positive and m^6^A-negative spike-ins are added to quantify m^6^A pull-down efficiency. After m^6^A enrichment, ERCC spike-ins are added to the input, supernatant, and eluent RNA pools as an internal standard for library preparation. The levels of m^6^A are quantified as the ratio of RNA abundance in eluent/(eluent + supernatant) [196].

#### 4.3.4. 5-Azacytidine-Mediated RNA Immunoprecipitation (Aza-IP-seq)

Another antibody-based sequencing technology utilizes the cytidine analog 5-azacytidine (5-azaC). Cells expressing RCMT or transfected with a tagged RCMT are incubated with the modified nucleoside, which is randomly incorporated into RNA. Due to the nitrogen substitution at the C5 position, when RCMT recognizes 5-azaC, it forms an irreversible covalent bond at the C6 position of its RNA targets, and therefore it cannot be released from the RNA [194]. IP is then performed using a specific antibody against RCMT or an anti-tag if an epitope-tagged RCMT is expressed in the cells, and the pulled-down RNAs are used for sequencing. The m^5^C sites are identified as C-to-G conversions, which results from a ring-opening of 5-azaC during the protocol [192]. One limitation of this method lies in the high toxicity of 5-azaC [197,198], permitting only a short treatment and thus reducing the probability of it being incorporated into the RNA.

#### 4.3.5. RNA Bisulfite Sequencing Technology (RNA-BisSeq)

RNA bisulfite sequencing technology (RNA-BisSeq) was initially used for detecting methylated cytosines in DNA [82]. At an acidic pH, sodium bisulfite reacts with cytosines, resulting in the deamination of unmethylated cytosines into uracil sulfonate; under a basic pH, this is converted to uracil, with the methylated cytosines remaining unchanged [192]. Therefore, this induces a C-to-U conversion of unmethylated cytosines, which can be detected by sequencing. RNA-BisSeq was not initially used for RNA methylation studies because the harsh conditions can induce RNA degradation. However, in 2009, Shaefer et al. detected methylated cytosines in tRNA and rRNA by lowering the denaturation temperature and extending the incubation time [199].

This approach has remarkable advantages that include single-nucleotide resolution and not requiring high concentrations of RNA. However, it fails to react with base-paired cytosines and cannot distinguish 5-methylcytosine from 5-hydroxymethylcytosine (hm^5^C) [192]. Additionally, some RNA secondary structures can prevent the C-to-U conversion, thus leading to incorrect identification of methylated cytosines [192,200,201].

#### 4.3.6. 2′-O-Methyl Sequencing (2′-O-Me-Seq)

2′-O-methyl sequencing (2′-O-Me-Seq) is a method used to map 2′-O-methyl sites in RNAs [202]. Under low dNTP concentrations, reverse transcription halts once it reaches a 2′-O-methylated nucleotide, thereby truncating the cDNA. The truncated cDNA is then sequenced to map the locations of 2′-O-methyl sites across the RNA sample. By using 2′-O-Me-Seq, researchers have been able to identify annotated 2′-O-methyl sites in rRNA and further identify 12 new sites [202].

#### 4.3.7. Ribose Methylation Sequencing (RiboMeth-Seq)

2′-O-methylated nucleotides can be detected with a sequencing method named RiboMeth-seq. 2′-O-methylated nucleotides are less sensitive to alkaline degradation when compared to unmethylated nucleotides [204]. RiboMeth-sequencing uses this property to detect 2′-O-methylations. The RNA is incubated at an alkaline pH and high temperature, allowing for its partial hydrolysis into 20–40 nucleotides fragments. The fragments are then ligated to adaptors using a tRNA ligase with no enzymatic activity, reverse transcribed, and sequenced [203].

The sequence is mapped to the reference sequence, and the first and last nucleotides of the library fragments sequence are recorded as 3′ and 5′. The nucleotides at the 3′ ends depend on their 2′OH function, while the 5′ ends depend on the 2′OH function of the neighbor fragment. The two reads are merged, and 5′ read ends are shifted one nucleotide upstream; thus, the reads refer to the same phosphodiester bond. Given that the 2′-O-Me nucleotide is resistant to degradation, it does not generate read ends, so the positions that correspond to a methylated nucleotide will be underrepresented. This creates a “negative image” that is converted to a peak diagram [203].

#### 4.3.8. Ribose Oxidation Sequencing (RibOxi-Seq)

This method starts with RNA fragmentation by the endonuclease Benzonase, which generates small RNA fragments; 2′-O-methylated nucleotides are resistant to fragmentation [205]. Oxidation/β-elimination is performed to remove the 3′ phosphates of fragmented RNA. 2′-O-methylation at the 3′ end renders the fragment resistant to oxidation, allowing the enrichment of 2′-O-methylation at the 3′ ends of fragmented RNAs by adaptor ligation [206]. The RNA fragments are further processed for sequencing. Finally, the terminal nucleotide of every fragment is recorded, and the processed data are referenced to a non-oxidized control [205]. The method requires microgram amounts of RNA, limiting the detection of low-abundance RNAs.

#### 4.3.9. Nm-Seq

Similar to RibOxi-seq, Nm sequencing is another method to detect 2′-O-methylation with base precision. Unlike RibOxi-seq, which relies on the random occurrence of fragmented 3′-end 2′-O-methyl, Nm-seq uses oxidation–elimination–dephosphorylation (OED) cycles to remove 3′-unmodified nucleotides and enrich RNA fragments with 3′ ends carrying 2′-O-methyl [71]. A final round of oxidation–elimination (OE) dephosphorylates any remaining 3′ end that does not contain 2′-O-methyl. The fragments ending with 2′-O-methyl undergo adapter ligation and then library construction for sequencing. Nm-seq can detect 2′-O-methyl in rRNA, mRNA, and ncRNAs.

#### 4.3.10. TAIL-Seq

TAIL-seq is a sequencing method that is designed for the sequencing of the 3′ poly-A tail of mRNA, can identify dynamic changes in the poly-A tail length, and can detect nucleotides added to the poly-A tail such as poly-U [127]. Prior to performing TAIL-seq, the sample RNA is depleted of rRNAs and small RNAs. A biotinylated 3′ adaptor is ligated to the RNA, and the RNA is partially fragmented with RNase T1. The 3′-end fragments are recovered via streptavidin pulldown. The resulting 3′ RNA is sequenced. Paired-end sequencing is used, where read 1 is used to identify the transcript, and read 2 is used to determine the 3′ poly-A tail. A specific algorithm is implemented to detect the signal intensity of long T stretches and other nucleotides at the 3′ end. TAIL-seq was the first method to detect widespread poly-U in short (~25 nt) poly-A tails [127].

#### 4.3.11. Borohydride Reduction (BoRed-Seq)

BoRed-seq is a method for detecting internal m^7^G which takes advantage of the nucleoside hydrolysis of m^7^G when treated with NaBH_4_ [119]. Total RNA is decapped to remove the 5′ m^7^G cap and treated with NaBH_4_ at a low pH to generate abasic sites at m^7^G nucleotides. The RNA is then treated with biotin-coupled aldehyde-reactive probe which will tag biotin to the abasic site. Streptavadin pulldown is used to enrich biotinylated RNA, which is sequenced. This method was used for the initial observation of m^7^G in miRNAs [119].

#### 4.3.12. Inosine Chemical Erasing Sequencing (ICE-seq)

Inosine chemical erasing sequencing (ICE-seq) is used to identify A-to-I editing sites in RNA [207]. This method employs a chemical reaction in which inosine is treated with acrylonitrile to form N^1^-cyanoethylinosine (ce^1^). Ce^1^ inhibits retrotranscription and truncates the cDNA at the site of RNA editing. The resulting cDNA is sequenced to identify sites of A-to-I editing. ICE-seq was initially used to identify A-to-I editing sites in human brain tissues [207].

#### 4.3.13. Detecting A-to-I Editing in RNAseq Data

Most epitranscriptomic modifications cannot be directly detected via RNAseq without alterations to the RNAseq protocol, as described above. When the RNA is reverse-transcribed into cDNA, the information of the modification site is lost [194]. A-to-I editing, however, can be detected directly in RNAseq data, as inosine is interpreted as a G during reverse transcription [209]. Detecting A-to-I editing requires specialized bioinformatics tools and sequencing depth to filter out small-nucleotide polymorphisms, false positives, and sequencing errors. Hoon et al. developed the first pipeline for detecting miRNA editing [210] from small RNA sequencing (sRNAseq) data. Lately, Alon et al. built a multi-step high-throughput sequencing strategy to systematically detect reliable canonical and non-canonical editing events from sRNAseq samples [211,212]. Since then, the pipeline has been refined to either allow the visualization of single mutations using MiRME [213] or detect editing along with miRNA isoforms using miRge 2.0 [214]. The bioinformatics tools are described in detail in Marceca et al. [208]. These tools were used to detect the A-to-I editing of miRNAs in tissue and plasma samples of NSCLC, highlighting the potential of edited miRNAs as a biomarker for lung cancer [169].

#### 4.3.14. Nanopore Sequencing

This approach was first used on RNA in 2018 by Gerald et al. [215], and it has been applied to the study of m^6^A, m^5^C, A-to-I editing, m^7^G, poly-U, and 2′-O-methyl [216,217,218,221,222,223,224]. It measures the RNA strand translocation into a nanopore protein inserted into the membrane. The estimation of the status of each nucleotide is based on the perturbation of the nanopore current, which is recorded when the RNA is translocated through the nanopore [215,217]. This method permits single-nucleotide resolution, does not require PCR amplification, and has the potential to detect a wide range of epitranscriptomic modifications. However, it shows a high signal-to-noise ratio and can fail to distinguish between nucleotides with similar structures [83]. Improvements to the base-calling algorithms and error corrections are being implemented to address these caveats [225]. A deviation of nanopore sequencing, nanopore-induced phase-shift sequencing (NIPSS), can be used to sequence small RNAs such as miRNAs [219]. NIPSS can distinguish 3′-end miRNA isoforms and modifications. Currently, NIPSS is limited to the first 14–15 nucleotides of the 3′ end of miRNAs and is unable to distinguish the 5′ seed region of a miRNA.

## 5. Concluding Remarks

In the past decade, there have been substantial advancements in the epitranscriptomics field due to the improvement of high-throughput methods for detecting and targeting epitranscriptomic modifications. These modifications can alter the functionality, structure, and regulation of their respective coding and non-coding RNAs, representing a novel, intricate regulation of the gene expression. In particular, miRNA regulation can be fine-tuned by epitranscriptomic modifications occurring on either the miRNA molecule or the targeted gene transcript. Several miRNAs that are well characterized in cancers can be regulated by epitranscriptomic modifications, modulating the biogenesis of these miRNAs or the efficiency of targets gene downregulation [9].

There is a wide range of regulatory networks under the control of epitranscriptomic modifications, many of which are involved in human physiology and pathology. Several of the enzymes involved in epitranscriptomics are suitable therapeutic targets, with m^6^A and A-to-I editing being the most studied of these modifications. Current pharmacological studies have identified small molecular inhibitors for METTL3 [226], showing promising results in inhibiting tumor progression and growth. ADAR1 is actively being studied as a potent therapeutic target for immuno-oncology therapy [227]. Other epitranscriptomic writers such as the NSUN family of proteins for m^5^C and HEN1 for 2′-O-methyl could be potential therapeutic targets for cancer. The poly-U enzymes TUT-4 and TUT-7 are being explored as potential therapeutic targets due to their role in let-7 miRNA maturation [228,229].

Another importance of these epitranscriptomic modifications in cancer lies in their possible detection in circulation as potential new-generation biomarkers, evolving a new, quick, inexpensive, and non-invasive method for cancer diagnosis. Recently, Ge et al. reported a significant upregulation of m^6^A levels in peripheral blood RNA of gastric cancer patients. Other studies have found elevated levels of m^6^A in the serum of cancer patients [230,231,232,233]. Decreased levels of m^5^C have also been found in the urine of colorectal cancer patients [166].

miR-17-5p shows increased methylation levels in cancer tissues compared to normal tissues, and its methylation level in serum could distinguish early pancreatic cancer patients from healthy patients [98]. Additionally, A-to-I editing in miRNAs has recently been used as a possible biomarker for cancer detection. A reduction in edited miRNAs has been reported in many human cancer tissues, resulting in an overexpression of their targets [166]. In 2018, Nigita et al. detected for the first time the deregulation of miRNA editing in circulating exosomes of lung cancer patients [169], laying the foundations for epitranscriptomic modifications as a possible biomarker for cancer. Notably, differential signatures in A-to-I miRNA editing have been described between White American and African American lung cancer patients, providing new profiling of canonical and modified miRNAs to study racial disparities in cancer [234]. The analysis of modified miRNAs in cancer permits a more comprehensive understanding of the mechanisms that drive the pathology and how these mechanisms are altered during cancer development and progression in different races. Studying the epitranscriptomic modifications in cancer biomarkers and their role in cancer pathogenesis may allow for a more tailored diagnosis and refine targeted therapeutic plans.

To add to this complexity, m^6^A itself can regulate or cooperatively interact with other epitranscriptomic modifications such as A-to-I editing, 2′-O-methylation, and m^5^C in RNAs. The interaction between m^6^A and A-to-I editing poses a possible mechanism whereby RNA editing can be fine-tuned by m^6^A regulation. The recently characterized METTL3 inhibitor could affect RNA editing activity, contributing to its effect on tumor cells. m^6^Am at the first nucleotide of an mRNA renders the mRNA resistant to miRNA downregulation. m^6^Am has its writer, CAPAM, and can be removed by the m^6^A eraser FTO [235]. It would be interesting to determine whether CAPAM is dysregulated in cancers and if oncogenes are hypermethylated with m^6^Am and vice versa for tumor suppressors. Future studies will need to consider these modifications to fully understand their scope in miRNA biology and their potential in human diseases.

## Figures and Tables

**Figure 1 genes-13-01289-f001:**
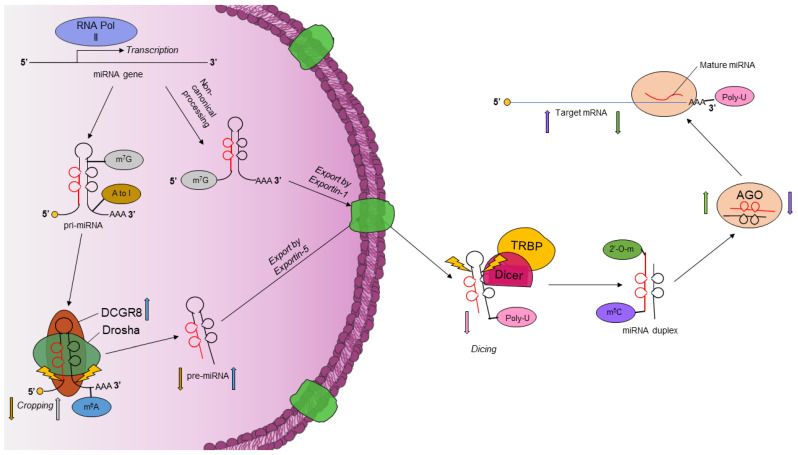
Epitranscriptomic modifications regulate the maturation and downstream targeting of miRNAs. A-to-I editing interferes with Drosha processing, inhibiting the processing of pri-miRNAs to pre-miRNAs (brown arrows). m^7^G (grey arrows) disrupts the formation of inhibitory G-quadruplexes in the pri-miRNA and facilitates miRNA processing. A group of m^7^G-capped miRNAs undergo a non-canonical biogenesis pathway, bypassing Drosha processing and being exported by exportin-1. m^6^A enhances DCGR8 binding to pri-miRNAs to enhance miRNA processing (blue arrows). m^5^C impairs mRNA/miRNA complex formation, affecting miRNA targeting (purple arrows). Poly-U blocks Dicer cleavage and marks the pre-miRNA for degradation (pink arrows). Poly-U is added to the miRNA-directed cleaved mRNA for 5′ degradation. 2′-O-methyl protects the 3′ end of miRNA from degradation and enhances AGO2 binding, increasing target repression by miRNAs (green arrows).

**Figure 2 genes-13-01289-f002:**
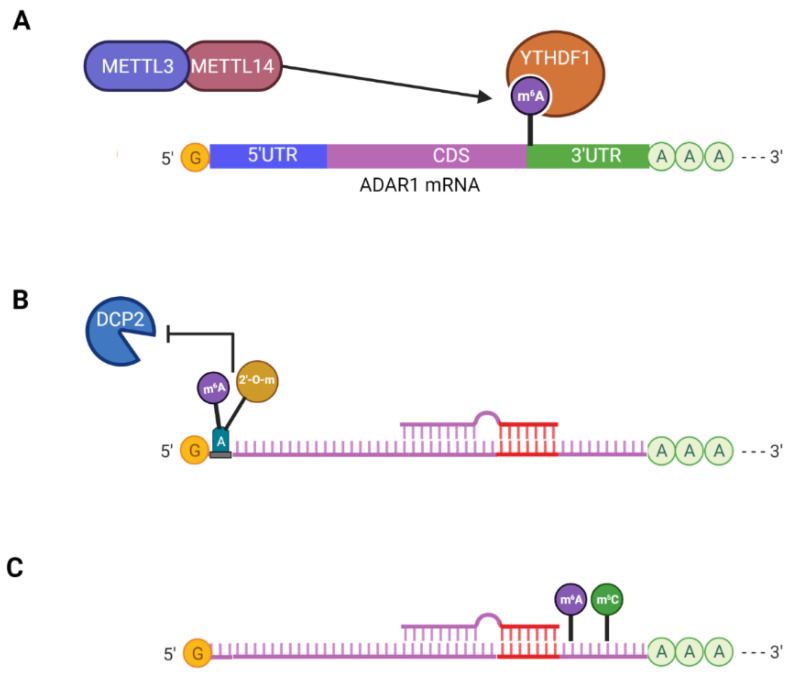
Crosstalk between epitranscriptomic modifications. (**A**) m^6^A regulation of ADAR1. The mRNA of ADAR1 is methylated by METTL3/METTL14 near the stop codon. This m^6^A mark recruits YTHDF1, which increases the protein translation of ADAR1. (**B**) m^6^Am modification at the first nucleotide. If the first nucleotide of an mRNA is adenine, the adenine can be methylated by m^6^A and 2′-O-methyl (m^6^Am). m^6^Am reduces the decapping activity of DCP2, thus rendering the mRNA resistant to miRNA-mediated degradation. (**C**) Cooperative interaction between m^6^A and m^5^C. m^6^A and m^5^C can cooperatively enhance the addition of each other to the 3′UTR of mRNAs. Both modifications occur close to the miRNA and AGO binding site. Their role in miRNA binding is speculative. Created with BioRender.com (accessed on 17 February 2022).

**Table 1 genes-13-01289-t001:** Techniques utilized for studying epitranscriptomic modifications.

Method	Specificity	Description	Advantages and Limitations	Suitability for miRNAs	References
Thin-layer chromatography	All modifications	Separates compounds in a mixture by their chemical properties. Each component migrates differentially based on affinity for the stationary (adherent) phase vs. mobile (liquid) phase. In 2D-TLC, the RNA is digested to form a 5′ OH prior to labeling with radioactive ATP. The migration is compared to a synthetic RNA standard, allowing for identification of specific epitranscriptomic modifications.	Can identify RNA modifications and be utilized for studying enzymatic activity and kinetics but does not provide the exact sites of the modifications.	Yes	[81,182,183,184]
Liquid chromatography–mass spectrometry (LC-MS)	All modifications	RNA samples are digested into nucleosides, which are separated into nucleotides by liquid chromatography, and the corresponding mass is determined by mass spectrometry. Using a ladder with known fragmentation patterns, the RNA sequence can be determined.	Can detect and quantify epitranscriptomic modifications with high sensitivity.	Yes	[185,186,187,188]
Methylated RNA immunoprecipitation coupled with high-throughput sequencing (MeRIP-seq), m^5^C-RIP-seq, and m^7^G-MeRIP	m^6^A, m^5^C m^7^G	Purified mRNA is randomly fragmented (~100–150 nucleotides) prior to immunoprecipitation with an anti-m^6^A antibody (MeRIP-seq), an anti-m^5^C (m^5^C-RIP-seq) antibody, or an anti m^7^G antibody (m^7^G-MeRIP). A library is constructed and sequenced.	High specificity but does not have single-nucleotide resolution and cannot detect methylation of non-abundant RNAs.	Yes	[7,22,82,111,189,190]
m^6^A-individual nucleotide resolution crosslinking and immunoprecipitation (miCLIP-m^6^A)	m^6^A	Implements UV crosslinking at the anti-m^6^A-bound site, which induces a mutation that can be identified by sequencing	Can identify the exact sites of m^6^A.	Not tested	[191]
m^5^C-individual nucleotide resolution crosslinking and immunoprecipitation (miCLIP-m^5^C)	m^5^C	A mutant of NSUN2 (C271A) is overexpressed which forms a covalent bond with m^5^C. The bond can be detected with an anti-NSUN2 antibody. This complex induces a stop position during RT-PCR, interpreted as a truncation site.	Can identify the exact sites of m^5^C.	Yes	[83,85,192]
Photo-crosslinking-assisted m^6^A sequencing (PA-m^6^A-seq)	m^6^A	Incorporates 4-thiouridine into the RNA, which induces a T-to-C mutation at crosslinked anti-m^6^A-bound sites that can be identified by sequencing.	Can identify the exact sites of m^6^A.	Not tested	[193,194,195]
m^6^A-level and isoform-characterization sequencing (m^6^A-LAIC-seq)	m^6^A	An excess of anti-m^6^A antibody is utilized for pulling down methylated RNA. A spike-in internal standard is added to allow for relative quantification of m^6^A RNAs	Permits evaluation of methylation status.	Not tested	[194,196]
5-azacytidine-mediated RNA immunoprecipitation (Aza-IP-seq)	m^5^C	5-azaC, a cytidine analog, is randomly incorporated into RNA. RCMT will form an irreversible bond with its RNA targets, which can be detected using an anti-RCMT antibody. m^5^C sites are recognized as C-to-G conversions due to a ring-opening of 5-azaC.	Can identify the exact sites of m^5^C, but only a short treatment is conducted due to the high toxicity of 5-azaC, thereby reducing its incorporation into RNA.	Not tested	[192,194,197,198]
RNA bisulfite sequencing technology (RNA-BisSeq)	Methylated cytosines such as m^5^C	Sodium bisulfite is added, which deaminates unmethylated cytosines (at acidic pHs) or uracil (at basic pHs), leaving methylated cytosines intact.	Has single-nucleotide resolution and does not use high concentrations of RNA. However, it cannot react with base-paired cytosines and does not distinguish 5-methylcytosine from 5-hydroxymethylcytosine	Yes	[82,192,199,200,201]
2′-O-methyl sequencing (2′-O-Me-Seq)	2′-O-methyl	Reverse transcription halts once it reaches a 2′-O-methylated nucleotide, thereby truncating the cDNA. These sites can be detected by sequencing.	Can detect specific 2′-O-methyl sites.	Not tested	[202]
RiboMeth-seq	2′-O-methyl	RNA is treated at an alkaline pH and high temperature to fragment the RNA into 20–40 nucleotides. The resulting fragments are sequenced. Sites that contain 2′-O-methyl sites are not fragmented and do not generate read ends.	Can detect omitted peak regions that corresponds to 2′-O-methyl sites.	Not tested	[203,204]
RibOxi-Seq	2′-O-methyl	RNAs are fragmented with Benzonase and oxidized to remove 3′ phosphates. 3′ ends that contain 2′-O-methyl are resistant to oxidation and are enriched with linker ligation.	Can detect 2′-O-methyl in rRNAs but requires microgram amounts of input.	No	[205,206]
Nm-seq	2′-O-methyl	Fragmented RNAs are treated with repeated cycles of OED, removing 3′ nucleotides that are not 2′-O-methylated. A final OE cycle is implemented to dephosphorylate non-2′-O-methylated 3′ ends, preventing adapter ligation.	Provides single nucleotide detection of 2′-O-methylation. Can be used for a wide range of RNAs.	Not tested	[71]
TAIL-seq	Poly-U	rRNA-depleted RNA samples are ligated in the 3′ end with a biotinylated adapter. RNA is fragmented with RNAse T1, and 3′ ends are recovered using streptavidin pulldown.	Provides information on poly-A tail length and the addition of poly-U at the 3′ end.	No	[127]
Borohydride Reduction (BoRed-seq)	m^7^G	RNA is decapped and treated with NaBH_4_ at a low pH. The abasic m^7^G site is treated with biotin-coupled aldehyde-reactive probe. The biotinylated RNA is recovered with streptavidin pulldown	Detects m^7^G site in RNAs without cleavage of the m^7^G sites. Suitable for small RNAs and low abundant RNAs	Yes	[119]
Inosine chemical erasing sequencing (ICE-seq)	A-to-I	Inosine is treated with acrylonitrile to form N1-cyanoethylinosinem, which halts retrotranscription and truncates the cDNA. These sites can be detected by sequencing.	Can identify A-to-I sites.	Yes	[207,208]
Bioinformatic detection of A-to-I editing from RNAseq	A-to-I	A-to-I editing is detected directly from RNAseq using bioinformatic tools to identify editing sites from SNPs	Can detect editing from RNAseq but requires high sequencing depth.	Yes	[169,194,208,209,210,211,212,213,214]
Nanopore sequencing	All modifications	Utilizes nanopore proteins that are inserted into the membrane. RNAs are translocated through these proteins, which leads to a perturbation of the nanopore current.	Has single-nucleotide resolution and does not require the processing of the amplified RNA. However, it has a high signal-to-noise ratio and may not distinguish similar nucleotides.	Yes	[83,215,216,217,218,219]

## Data Availability

No new data were created or analyzed in this study. Data sharing is not applicable to this article.

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
