# Peer review of "The Epitranscriptome in miRNAs: Crosstalk, Detection, and Function in Cancer"

_genes, 2022, doi:10.3390/genes13071289_

Round 1
Reviewer 1 Report
In this manuscript, Acunzo and colleagues systematically summarized current knowledge on the RNA modifications occurring on miRNAs, including function, detection technologies and crosstalk. Most contents are well organized and the language is easy to follow. However, I strongly recommend that the following comments are completely addressed in the revised manuscript.
1. 3’ RNA Uridylation and m7G methylation have also been identified in miRNA and play important roles in miRNA biogenesis and function. Hence, I strongly recommend the authors to describe and summarize the biological functions of these two modifications in miRNA.
2. In section 4 “Methods for detecting epitranscriptomic modifications”, since all the methods have both advantages and disadvantages, the authors should describe or point out which methods are suitable for RNA modifications detection in miRNA.
3. For 2’-O-methyl sequencing, other sequencing methods, including RiboMeth-seq, RibOxi-seq, Nm-seq, MeTH-seq, should also be summarized.
4. In L227, the title should be “m5C in miRNAs” but not “2’-O-methyl in miRNAs”
5. In L371, the reference “Cap-specific terminal N6-methylation of RNA by an RNA polymerase II-associated methyltransferase” is omitted.
Author Response
"Please see attachment below"

Reviewer 2 Report
In the review “The epitranscriptome in miRNAs: crosstalk, detection and function in cancer”, del Valle-Morales et al. address the main epitranscriptomics marks present in miRNAs: m6A, m5C, 2’-O-me and A-to-I editing. This review follows some other recent publications that discuss the role of epitranscriptomic marks in small RNAs, but adds a very interesting revision of the crosstalk between different modifications. Manuscript is well cited and could be interesting for the readers of this journal.
Abstract and introduction are very well written and properly introduce the reader to the topic. Section 2 is well written and informative, but some sections have pieces of key information missing. Section 3 "crosstalk between epitranscriptomic modifications" is very interesting and open new questions about the crosstalk between epitrasncriptomic marks in miRNAs. In section 4 "Methods for detections epitranscriptomics modifications" in some parts authors focus on m5C and m6A completely ignoring that some techniques are also used for many other modifications, including the other two that this manuscript specifically address.
Comments:
- Apart of 3’ UTR of mRNAs, m6A has been found to be highly enriched near stop codons, within long internal exons, in intergenic regions, introns, and at 5’UTRs.
- Many important cofactors of m6A methylation complex (VIRMA, ZC3H13 and HAKAI) and other m6A methyltransferases (METTL5, METTL16) are missing.
- Recently, a very important observation regarding the role of FTO as m6A demethylase has been made (doi: 10.1038/nature21022). This article experimentally demonstrates that FTO demethylate m6Am rather than m6A, which is a very important finding in the field. This article is cited in the review regarding the crosstalk between m6A and 2’-O-me, but this important finding has been ignored. The controversial role of the m6A demethylase FTO should be addressed in the m6A section of this review.
- In the section “m5C methylation”, DNA methyltransferase 2 (DNMT2), that methylates tRNAs, should be added when addressing m5C methyltransferases.
- In lines 202-203, it seems that authors name NSUN2 as “NOL1/NOP2/SUN domain”. That is the name of the family, not NSUN2. The name of the family should be stated at line 196, when the family is mentioned for the first time. Official complete name of NSUN2 is NOP2/Sun RNA methyltransferase 2 according to NCBI.
- Section 2.3.1. has the wrong title.
- It is not clear how the sentence that starts at line 328 and finishes at line 330 (“However, specific methodologies, such as Sanger sequencing and single-base primer extension (SNaPshot) assay, are available to detect edited miRNAs and their editing sites”) is related with the rest of the paragraph.
- I have some concerns about section 4.3.2. Authors describe UV crosslinking methods making them look as if they were specific for m6A and m5C, but this method can be used for any RNA binding protein (there are several examples of iCLIP or PAR-CLIP performed for different methyltransferases and different modifications, miCLIP is just an modification of iCLIP original protocol). This can be confusing for the reader. I suggest that authors change this section clarifying that UV crosslinking methods can be used for different modifications and different RBP. This should be also changed in table 1. In addition, miCLIP for NSUN2 does not require UV crosslinking when using mutant C271A. Mutation of Cysteine 271 to Alanine generates an enzyme that is unable to release the RNA after the modification. Thus, the RNA is covalently bound to the enzyme, and no UV crosslinking is required. This should be also corrected in table 1.
- In section 4.3.4. authors indicate that for Aza-IP-seq is necessary to overexpress the RCMT but that is not true. Aza-IP-seq requires either a very good antibody against the methyltransferase to immunoprecipitated the endogenous protein or the expression of a tagged version of the enzyme. But overexpression of the RCMT is not strictly necessary and should not be stated as fact.
- In section 4.3.9. authors say that nanopore sequencing is used to study m6A and m5C, but nanopore is not only limited to these two modifications. Among other many modifications, A-to-I editing, 2'-O-methyl, pseudouridine, ac4C or m2,6A have also been detected by nanopore sequencing (https://doi.org/10.1038/s41592-022-01513-3 ; https://doi.org/10.7554/eLife.76562 https://doi.org/10.1016/j.xgen.2022.100097). This should be corrected as it could be confusing for the reader.
Author Response
"Please see the attachment."
